# Unsupervised Domain Adaptation for Semantic Segmentation using Depth Distribution

**Quanliang Wu, Huajun Liu**[*]
School of Computer Science
Wuhan University, Wuhan, China
{quanliangwu, huajunliu}@whu.edu.cn

## Abstract

Recent years have witnessed significant advancements made in the field of unsupervised domain adaptation for semantic segmentation. Depth information has been proved to be effective in building a bridge between synthetic datasets and real-world datasets. However, the existing methods may not pay enough attention to depth distribution in different categories, which makes it possible to use them for further improvement. Besides the existing methods that only use depth regression as an auxiliary task, we propose to use depth distribution density to further support semantic segmentation. Therefore, considering the relationship among depth distribution density, depth and semantic segmentation, we propose a branch balance loss for these three sub-tasks in multi-task learning schemes. In addition, we also propose a spatial aggregation priors of pixels in different categories, which can be used to refine the pseudo-labels for self-training, thus further improving the performance of the prediction model. Experiments on SYNTHIA-to-Cityscapes and SYNTHIA-to-Mapillary benchmarks show the effectiveness of the method. The source code is available at https://github.com/depdis/Depth_Distribution.

## 1 Introduction

Semantic segmentation refers to the task of assigning semantic categories to each pixel in an image, such as sky, road, car, etc. It is very important for many applications, such as autonomous driving [1] and image editing [2]. In this field, approaches of recent years are mostly based on fully convolutional network (FCN) [3] with modifications designed for pixel-wise prediction [4, 5, 6]. To tackle the challenge of large-scale annotations, unsupervised domain adaptation (UDA) is broadly used. A typical practice of UDA is to adapt the semantic segmentation model trained on synthetic datasets [7] (source domain) to perform on real-world datasets [8, 9] (target domain). In order to obtain better adaptation, the GAN [10] structure is widely used to minimize the feature distribution discrepancy, so that the model can utilize the knowledge learned from the source domain and apply it to the target domain.

As depth information has been proved to be effective in promoting the performance of semantic segmentation [11, 12, 13, 14], it is usually used to help build the semantic connection between two domains. SPIGAN [11] makes use of the privileged information in the source data from a simulator through the privileged network, which serves as an auxiliary task and is regularized to the main segmentation network. DADA [12] is a depth-aware domain adaptation scheme to dig deeper into geometry information, which not only executes the depth regression task when training the generator, but also fuses it together with semantic information during adversarial learning. CTRL [13] uses multi-task learning to establish the relationships between different visual semantic categories and depth levels in UDA context. On the premise of obtaining or generating depth information in advance,

---

[*]Corresponding author.

CorDA [14] leverages self-supervised depth estimation to bridge the domain gap. And the correlations between semantics and depth greatly improves the performance of target semantic segmentation in the presence of a domain shift. However, the way of using depth information needs to be further explored.

[13] mentions that different semantic categories have discrete depth value ranges, but we find that different semantic classes also have their own depth distribution in images. For example, the sky is at the top of the image, the road is at the bottom of the image, cars and buses are driving on the road, people are always walking on the sidewalk, and so on. According to the above observation, different semantic categories might display relatively consistent depth distribution at their positions in images. Therefore, we try to make use of the structural information in a more accurate way. In this paper, we propose to use the Gaussian mixture models(GMMs) to build the depth distribution for different semantic classes, which can give an estimate of the probabilities of both in-category and out-of-category pixels within the UDA context for semantic segmentation. Concretely, looking into a certain category, such as car, the depth distribution of their locations can be learned by a mixture of several Gaussian models, which share common parameters across images and domains. In order to add this information to semantic segmentation, we use the multi-task learning framework to learn three sub-tasks, i.e. semantic segmentation prediction, depth regression and depth distribution density estimation. In addition, considering the internal relations among three sub-tasks, we propose to use density estimation to balance the other two sub-tasks, mainly to improve the efficiency of the main task(semantic segmentation). Moreover, we use the total proportion of class-wise pixels in the source images as a prior to setting different thresholds, so as to refine pseudo-labels and improve the efficiency of self-supervised learning on the target domain. We have demonstrated the effectiveness of our proposed method on the benchmark tasks SYNTHIA-to-Cityscapes and SYNTHIA-to-Mapillary, on which we have obtained new state-of-the-art segmentation performance.

The main technical contributions of our work are made possible as follows:

- We propose to utilize the probability density of depth distribution, which is similar in source domain and target domain, to bridge the domain gap in UDA for semantic segmentation in a more accurate way.

- Based on the proposed density branch, an idea of branch balance training is proposed for our multi-task learning, and a branch balance strategy is designed to promote segmentation performance.

- A pseudo-labels refinement algorithm is proposed, which uses the aggregation priors of pixels in different categories on the source samples.

## 2 Related Work

**Unsupervised domain adaptation.** A typical practice of UDA is to adapt a semantic segmentation model trained on synthetic datasets (source domain) to perform on real-world datasets (target domain) [15, 16, 17, 18, 19, 20, 21, 22, 23, 24]. Hoffman and colleges are the first to leverage adversarial training to conduct domain adaptation in the semantic segmentation tasks, where features are aligned and the label statistics of the source domain are transferred by category specific adaptation [25]. Then, other methods are to align the distribution of source and target domains through output space [26] or feature space [18, 19], align input pixels of the source and target images [27, 28], or to refine pseudo-labels under the self-training framework [17, 29, 30, 31, 32]. The above methods mainly focus on improving the architecture of the segmentation networks, but may not make use of the rich structure information of the two domains.

**Use of geometric information in semantic segmentation.** To further boost the domain adaptation in semantic segmentation, some researches have explored the use of depth information in the source data, which is the additional information available only during training time[11, 12, 33, 13, 14]. The depth information has been increasingly used to help the domain adaptation, but the relationship between depth and semantic segmentation needs to be further explored in a more accurate way. Our method is similar to DADA[12] and CTRL[13], but we use GMMs to learn the depth distribution possibilities for different categories in the images, so as to establish the relationship between synthetic data and real data more accurately.

**Multi-task learning.** Our work makes use of multi-task learning (MTL) [34], where multiple tasks are predicted by different branches. SPIGAN[11], DADA[12], GIO-Ada[33], CTRL[13] all use the

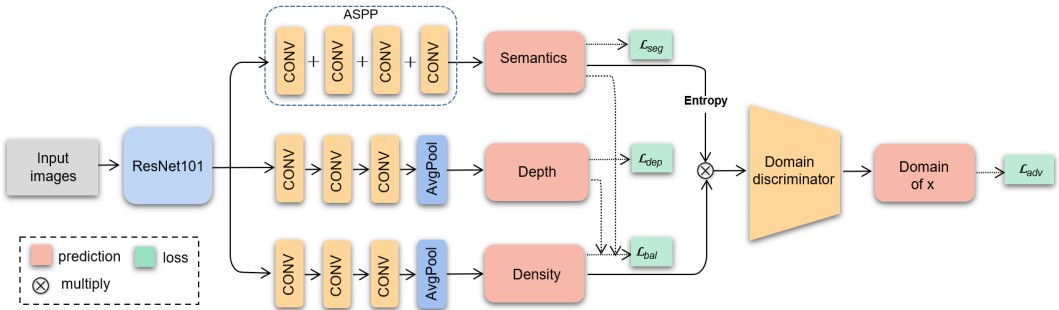

Figure 1: An overview of our network. The backbone is Resnet101, which follows three branches. In the main task branch, the features of the backbone CNN are fed into Deeplab-V2 [6] architectural design with Atrous Spatial Pyramid Pooling (ASPP) for semantic segmentation prediction. In the depth regression branch, the backbone CNN features are continuously forwarded to three convolutional blocks, followed by an average pooling layer, to output depth. We estimate the depth distribution density using the same network as the depth regression branch. We fuse the features extracted by density estimation and the features extracted by semantic segmentation for adversarial training.

MTL for depth regression and semantic segmentation, and make use of the depth information to bridge the domain gap. However, apart from these two tasks, no other new tasks have been proposed to help semantics. Although CTRL adds a refined segmentation branch as the third task, there is no close connection between these tasks. We use the standard multi-task learning, and add a depth distribution density estimation branch to predict a density map which is closely associated with depth and semantic segmentation. Therefore, we design a branch balance loss for the three branches, which can significantly improve the accuracy of segmentation. Compared to CTRL, our method has a simpler network and better prediction performance.

**Pseudo-label refinement under self-training frameworks.** Our work is also related to self-supervised learning. By iteratively training the network with gradually improved target pseudo-labels, the performance on the target domain can be further improved[17, 23, 20, 13]. Different from their approaches, we explore and add the spatial aggregation priors in different categories of source domain, as a guide to refine the pseudo-labels for self-training.

## 3 Methodology

A fitted GMM can give an estimate of the probabilities of both in-sample and out-of-sample data points, which is known as density estimation. On the observation that the depth distribution of the same category between synthetic samples and real samples is similar[1], we can utilize this information from source samples and transfer it to target samples using adversarial training. We use standard multi-task learning framework to obtain three sub-tasks (as shown in Fig. 1), namely semantic segmentation (main task), depth regression and depth distribution density estimation. In our framework, two auxiliary tasks work together to improve the performance of the main task. Following an adversarial training scheme[25, 26], we train the network in a UDA setting. In addition, we also explore pixel aggregation priors of different categories on the source domain to help refine the pseudo-labels on the target domain, thus further improving the semantic segmentation performance.

### 3.1 Problem Formulation

Let $\mathfrak{D}_s$ and $\mathfrak{D}_t$ represent the source domain and target domain, and the samples from them are represented by tuples $(I_s, P_s, Z_s)$ and $(I_t)$ respectively, where $I \in \mathbb{R}^{H \times W \times 3}$ are color images, $P \in \{1, ...C\}^{H \times W}$ are semantic annotations with $C$ classes, and $Z \in [Z_{min}, Z_{max}]^{H \times W}$ are depth maps from a finite frustum. For the source domain training, we optimize the network cost using supervised loss for semantic segmentation and depth regression. Therefore, an unsupervised adversarial learning scheme on the target domain is within the same training process. In addition, we

---

[1]Please see the GMM parameters file of SYNTHIA and Cityscape datasets on the source code website.

use a branch balance loss to estimate the depth distribution density on both source and target domains, which is closely related to segmentation and depth results.

## 3.2 Supervised Learning

For **semantic segmentation prediction**, we adopt the standard cross-entropy loss, which is represented as pixel-wise class probabilities over $C$ classes:

$$\mathcal{L}_{seg}(\hat{P}, P) = -\sum_{i=1}^{C} P_i \log \hat{P}_i, \tag{1}$$

where $\hat{P}$ and $P$ are the predicted and ground truth segmentation maps, respectively.

For **depth regression**, we employ the berHu loss:

$$\mathcal{L}_{dep}(\hat{Z}, Z) = berHu\left(\hat{Z} - Z\right), \tag{2}$$

with the reverse Huber loss defined as [35]:

$$berHu\left(e_z\right) = \begin{cases} |e_z|, & |e_z| \leq L \\ \frac{(e_z)^2 + L^2}{2L}, & |e_z| > L \end{cases} \tag{3}$$

$$L = 0.2 max\left(|e_z|\right), \tag{4}$$

where $\hat{Z}$ and $Z$ are the predicted and ground truth depth maps, respectively. Following [33, 12], inverse depth is adopted for the depth regression loss.

## 3.3 Depth Distribution Density Estimation

For **density estimation**, a density map of each image should be built first. Since the ground truth segmentation and depth information of source samples have been provided, this information can be used to construct Gaussian mixture models for each semantic category.

For each category $i \in (1, ..., C)$, the depth value $z$ of the pixel associated with its position $(x, y)$ can form a tuple $(x, y, z)$, and the list of tuples $(x, y, z)$ where the associated pixel is classified as $i$th class can be denoted as $\vec{X}_i$. For each category $i$, we adopt Expectation-Maximization algorithm [36] to learn the mixture component weights $\phi_{ij}$, the component means $\vec{\mu_{ij}}$ and variances/covariances $\Sigma_{ij}$ of GMMs from the depth distribution information $\vec{X}_i$ on source samples, where $j$ means the $j$th component of a GMM. Therefore, we can learn the GMMs for each semantic class on the source domain.

The density values of each pixel can be calculated by the following formula:

$$p\left(\vec{X}_i\right) = \sum_{j=1}^{K} \phi_{ij} \mathcal{N}\left(\vec{X}_i \mid \vec{\mu_{ij}}, \Sigma_{ij}\right), \tag{5}$$

where $\sum_{j=1}^{K} \phi_{ij} = 1$, and $\vec{X}_i \sim \mathcal{N}(\vec{\mu_{ij}}, \Sigma_{ij})$. In our experiment, $K$ is defined as 5.

By calculating the density value of all the pixels by class-wise GMMs, a depth distribution density map $D \in \mathbb{R}^{H \times W}$ of the image is constructed, which can be regarded as the ground truth. For Equation 5, every pixel in $D$ describes the probability of seeing the depth value with this semantic class at this pixel location under the learned GMM.

We propose a **branch balance loss** for the depth distribution density regression, and its novelty lies in the different construction ways of the density map $D$. For *source domain training*, instead of using the pre-calculated density maps of the source samples, we use the ground truth depth, the predicted segmentation map and pre-constructed source domain GMMs to generate the density map $D_s$ for each sample. For *target domain training*, we use the estimated depth, the predicted segmentation maps and pre-constructed source domain GMMs to generate the density map $D_t$ for each sample. We also use berHu loss (the reversed Huber criterion [35]) to estimate the density. Therefore, the branch balance loss can be expressed as:

$$\mathcal{L}_{bal}(\hat{D}, D) = berHu\left(\hat{D} - D\right), \tag{6}$$

where $\hat{D}$ is the predicted density map of the density branch, and $D$ is the constructed density map by the other two branches.

Note that, unlike the losses used for semantic segmentation prediction and depth regression, the branch balance loss is suitable for the training of source domain and target domain. There are two advantages to the proposed branch balance loss. On one hand, it can balance the training efficiency of different branches on the source domain and target domain, especially on the target domain. Because there exists no ground truth to constrain the sub-tasks training on the target domain, it may not be easy for each branch to keep balance. In addition, because we use the GMMs of the source domain to calculate density maps of the target samples, we also build a bridge between the two domains.

The network parameters denoted as $\theta_{net}$ are learned to minimize the following objective functions on the source domain and target domain:

$$\min_{\theta_{net}} \mathbb{E}_{\mathfrak{D}^{(s)}} \left( \lambda_{seg} \mathcal{L}_{seg} + \lambda_{dep} \mathcal{L}_{dep} + \lambda_{bal} \mathcal{L}_{bal} \right), \tag{7}$$

$$\min_{\theta_{net}} \mathbb{E}_{\mathfrak{D}^{(t)}} \left( \lambda_{tar} \mathcal{L}_{bal} \right), \tag{8}$$

where hyperparameters $\lambda_{seg}$, $\lambda_{dep}$, $\lambda_{bal}$ and $\lambda_{tar}$ are the weight values. In our experiments, we use $\lambda_{seg} = 1.0$, $\lambda_{dep} = 0.5 \times 10^{-2}$, $\lambda_{bal} = 10^{-2}$, $\lambda_{tar} = 5 \times 10^{-2}$.

## 3.4 Adversarial Training

We use a discriminator network to align the source and target domains. Unlike CTRL[13] which designs a Cross-Task Relation Layer to concatenate three entropy maps (semantic segmentation, refined semantic segmentation and depth) together, we follow the DADA fusion strategy. However, different from DADA, we use segmentation to fuse density map rather than depth map.

To be more specific, the weighted self-information maps [19] $F\left(\hat{P}\right)$ is calculated at first by:

$$F\left(\hat{P}\right) = -\hat{P} \odot \log \hat{P}. \tag{9}$$

Then, we fuse the $C$-channel $F$ with the estimated density map $\hat{D}$ to produce a fused $C$-channel map $\hat{F}$, and feed it forward to a discriminator.

More specifically, the discriminator network parameters $\theta_{\mathcal{D}}$ can be obtained by correctly classifying the sample domain as a source or target during training:

$$\min_{\theta_{\mathcal{D}}} \left\{ \mathbb{E}_{\mathfrak{D}_s} \left[ \log \mathcal{D}\left(\hat{F}_s\right) \right] + \mathbb{E}_{\mathfrak{D}_t} \left[ \log \left( 1 - \mathcal{D}\left(\hat{F}_t\right) \right) \right] \right\}. \tag{10}$$

At the same time, the prediction network parameters $\theta_{net}$ is updated using the "fooling" objective minimization:

$$\min_{\theta_{net}} \mathbb{E}_{\mathfrak{D}_t} \left[ \log \mathcal{D}\left(\hat{F}_t\right) \right]. \tag{11}$$

The hyperparameter $\lambda_{adv}$ is used to weigh the relative importance of the adversarial loss (Equation 11), and we set $\lambda_{adv} = 5 \times 10^{-2}$ in our experiment. In our training scheme, the model parameters of the prediction network ($\theta_{net}$) and the discriminator ($\theta_D$) are optimized jointly. At each training iteration, we input a batch of two samples from the source domain and the target domain into the network. All loss gradients are accumulated and then propagated back to update the network.

## 3.5 Spatial Aggregation Priors for Pseudo-labels Refinement

After multi-task learning in the framework of adversarial training, self-training can further improve the effect of semantic segmentation. We find that pixels of large objects, such as sky and road, have a large-scale aggregation in image space, while pixels of small objects, such as person and bicycle, have relatively small-scale aggregation in image space. Therefore, since most pixels in the same class are clustered together, we calculate the total proportion of class-wise pixels in the source images as a prior, to set different thresholds for refining pseudo-labels on the target domain for self-training.

$$thres_i = N_{base0} + \frac{N_i - N_{min}}{N_{max} - N_{min}} \times N_{base1}, \tag{12}$$

where $thres_i$ represents the threshold of different category $i$, $N_{min}$ and $N_{max}$ represent the minimum and maximum numbers of pixels of different categories from all source samples, respectively. $N_i$ represents the number of pixels of the $i$th category from all source samples. $N_{base0}$ and $N_{base1}$ are two default values, which are set as 50 and 4950 respectively in our experiment, so the threshold ranges from 50 to 5000. Please note that, because the aggregation scale of different categories varies greatly, we set a threshold range of 1:100, which can be further adjusted according to the actual size of classes in different datasets.

We design an algorithm which adds spatial aggregation priors for refining pseudo-labels. Algorithm 1 is the semantic description of refining the pseudo-labels by using the spatial aggregation priors.

---

**Algorithm 1: Spatial prior pseudo-labels refinement algorithm**

---

**Input:**    A target sample with predicted pseudo-labels.
**Output:**   Refined pseudo-labels.
1 Initialize all pixels to set their flags $T_{wh}$=0.
2 **for** $w$=0 to $W$ **do**
3     **for** $h$=0 to $H$ **do**
4         **if** $T_{wh}$=0 && $Confidence_{wh}$ ≥0.9 **then**
5             Search around it for pixels that satisfy the following conditions:
6                 Their prediction class is the same as $T_{wh}$, and their confidence value ≥ 0.9.
7                 Iterate over taking these points as the fiducial points and search around them outward for the qualified points.
8         Count the number of all qualified pixels, and record as $N_c$;
9         **if** $N_c \geq thres_i$ **then**
10           Set flags of all these pixels to 1;
11 Pixels labeled with 1 are reserved, and their pseudo-labels can be used for self-supervised learning.

---

Our prediction model can be further improved by self-training schemes. Follow [17], we first train the prediction and discriminator networks for Q1 iterations. We generate semantic pseudo-labels on the target training samples using the trained prediction network, and then obtain refined pseudo-labels by Algorithm 1 to further train the prediction network on the target training samples using pseudo-labels supervision for Q2 iterations. The above pseudo-labels generation-refinement and self-training process is executed twice to produce higher quality semantics output on the target domain.

# 4 Experiments

## 4.1 UDA Benchmarks

To make a fair comparison with previous methods[11, 26, 37, 38, 19, 12, 13], especially DADA and CTRL, which are the most similar to our method, we use three standard UDA evaluation protocols to verify our model: SYNTHIA → Cityscapes (16 classes), SYNTHIA → Cityscapes (7 classes), and SYNTHIA → Mapillary (7 classes). Detailed descriptions of these settings can be found in [12] and [13]. In all settings, the SYNTHIA dataset [7] is used as the source domain. Following [12, 13], we use the SYNTHIA-RAND-CITYSCAPES split consisting of 9,400 synthetic images and their corresponding pixel-wise semantic labels and depth. For target domains, we use Cityscapes [8] and Mapillary Vistas [9] datasets. Similar to [26, 19, 12, 13], we report the performance of semantic segmentation based on "mean Intersection over Union" (mIoU in %) on the 16 classes of the Cityscapes validation set, and also show the mIoU (%) of the 13 classes (mIoU*) excluding classes with *. For depth, we use Absolute Relative Difference ($|Rel|$), Squared Relative Difference ($Rel^2$), Root Mean Squared Error ($RMS$), its log-variant $LRMS$; and the accuracy metrics [39] as denoted by $\delta_1$, $\delta_2$, and $\delta_3$. For each metric, we use ↑ and ↓ to denote the improvement direction.

## 4.2 Experimental Setup

All our experiments are conducted on a single NVIDIA 1080Ti GPU with a memory of 11GB. Our network is implemented on PyTorch [40]. Backbone is a ResNet-101 [41] initialized with ImageNet [42] weights. Like [26, 19], we apply Atrous Spatial Pyramid Pooling (ASPP) with sampling rates of {6, 12, 18, 24}. In addition, we use DC-GAN [43] as our domain discriminator for adversarial

Table 1: The quantitative results of different methods for semantic segmentation performance (IoU and mIoU, %) on SYNTHIA→ Cityscapes(16 classes).

| Models | Depth | road | sidewalk | building | wall* | fence* | pole* | light | sign | veg | sky | person | rider | car | bus | mbike | bike | mIoU↑ | mIoU*↑ |
|---|---|---|---|---|---|---|---|---|---|---|---|---|---|---|---|---|---|---|---|
| SPIGAN[11] | √ | 71.1 | 29.8 | 71.4 | 3.7 | 0.3 | **33.2** | 6.4 | 15.6 | 81.2 | 78.9 | 52.7 | 13.1 | 75.9 | 25.5 | 10.0 | 20.5 | 36.8 | 42.4 |
| AdaptSegnet[26] | | 79.2 | 37.2 | 78.8 | – | – | – | 9.9 | 10.5 | 78.2 | 80.5 | 53.5 | 19.6 | 67.0 | 29.5 | 21.6 | 31.3 | – | 45.9 |
| AdaptPatch[37] | | 82.2 | 39.4 | 79.4 | – | – | – | 6.5 | 10.8 | 77.8 | 82.0 | 54.9 | 21.1 | 67.7 | 30.7 | 17.8 | 32.2 | – | 46.3 |
| CLAN[38] | | 81.3 | 37.0 | 80.1 | – | – | – | 16.1 | 13.7 | 78.2 | 81.5 | 53.4 | 21.2 | 73.0 | 32.9 | 22.6 | 30.7 | – | 47.8 |
| Advent[19] | | 87.0 | 44.1 | 79.7 | 9.6 | 0.6 | 24.3 | 4.8 | 7.2 | 80.1 | 83.6 | 56.4 | 23.7 | 72.7 | 32.6 | 12.8 | 33.7 | 40.8 | 47.6 |
| DADA[12] | √ | **89.2** | **44.8** | **81.4** | 6.8 | 0.3 | 26.2 | 8.6 | 11.1 | 81.8 | **84.0** | 54.7 | 19.3 | 79.7 | 40.7 | 14.0 | 38.8 | 42.6 | 49.8 |
| CTRL[13] | √ | 86.9 | 43.0 | 80.7 | 19.2 | 0.9 | 27.2 | 11.6 | 12.6 | 81.3 | 83.2 | 60.7 | 24.0 | 84.2 | 46.2 | 22.0 | 44.2 | 45.5 | 52.4 |
| Ours | √ | 85.3 | 40.2 | 79.7 | **19.6** | **1.3** | 29.4 | **29.7** | **32.2** | 82.5 | 79.2 | **64.3** | **26.7** | **85.2** | **49.4** | **22.7** | **44.9** | **48.2** | **55.5** |

Table 2: The quantitative results of different methods for semantic segmentation performance (IoU and mIoU, %) on SYNTHIA→ Cityscapes(7 classes) and SYNTHIA → Mapillary (7 classes) in low-resolution and full-resolution.

| Res. | Model | Depth | (a) SYNTHIA → Cityscapes (7 classes) | | | | | | | | (b) SYNTHIA → Mapillary (7 classes) | | | | | | | |
|---|---|---|---|---|---|---|---|---|---|---|---|---|---|---|---|---|---|---|
| | | | flat | const | object | nature | sky | human | vehicle | mIoU↑ | flat | const | object | nature | sky | human | vehicle | mIoU↑ |
| 320*640 | SPIGAN[11] | √ | 91.2 | 66.4 | 9.6 | 56.8 | 71.5 | 17.7 | 60.3 | 53.4 | 74.1 | 47.1 | 6.8 | 43.3 | 83.7 | 11.2 | 42.2 | 44.1 |
| | Advent[19] | | 86.3 | 72.7 | 12.0 | 70.4 | 81.2 | 29.8 | 62.9 | 59.4 | 82.7 | 51.8 | 18.4 | 67.8 | 79.5 | 22.7 | 54.9 | 54.0 |
| | DADA[12] | √ | 89.6 | 76.0 | 16.3 | 74.4 | 78.3 | 43.8 | 65.7 | 63.4 | 83.8 | 53.7 | **20.5** | 62.1 | 84.5 | 26.6 | 59.2 | 55.8 |
| | CTRL[13] | √ | 90.8 | 77.5 | 15.7 | 77.1 | **82.9** | 45.3 | 68.6 | 65.4 | **86.6** | 57.4 | 19.7 | **73.0** | **87.5** | **45.1** | **68.1** | **62.5** |
| | Ours | √ | **92.6** | **78.2** | **23.4** | **77.2** | **82.9** | **49.6** | **69.8** | **67.7** | 86.2 | **58.7** | 19.4 | 68.9 | 86.1 | 40.4 | 62.4 | 60.3 |
| Full | Advent[19] | | 89.6 | 77.8 | 22.1 | 76.3 | 81.4 | 54.7 | 68.7 | 67.2 | 86.9 | 58.8 | 30.5 | 74.1 | 85.1 | 48.3 | 72.5 | 65.2 |
| | DADA[12] | √ | 92.3 | 78.3 | 25.0 | 75.5 | 82.2 | 58.7 | 72.4 | 70.4 | 86.7 | 62.1 | **34.9** | 75.9 | 88.6 | 51.1 | 73.8 | 67.6 |
| | CTRL[13] | √ | **92.4** | 80.7 | 27.7 | 78.1 | **83.6** | 59.0 | **78.6** | 71.4 | **88.5** | 59.2 | 27.8 | **79.4** | 85.7 | **64.4** | **79.6** | 69.2 |
| | Ours | √ | **92.4** | **81.8** | **34.3** | **78.9** | 82.0 | 64.5 | 74.1 | **72.6** | 87.7 | **68.6** | 33.7 | 74.8 | **93.0** | 61.4 | 73.4 | **70.4** |

learning. The learning rates of the prediction and discriminator networks are set as $2.5 \times 10^{-4}$ and $1.0 \times 10^{-3}$ respectively. In self-training, the parameters are: Q1 = $54K$, Q2 = $30K$.

## 4.3 Comparison to Other Methods

### 4.3.1 SYNTHIA → Cityscapes(16 classes)

Table 1 reports semantic segmentation performance of our proposed model on SYNTHIA → Cityscapes(16 classes). It shows our method achieves SOTA performance on both 16 and 13 classes, outperforming other methods by large margins. Compared to the SOTA works, we have improvements in 11 classes. Compared with the SOTA CTRL[13], which maps depth to a discrete probability space, our main gains come from the following classes – "light" (+18.1%), "sign" (+19.6%), "person" (+3.6%), "bus" (+3.2%), "rider" (+2.7%) and "pole" (+2.2%). Besides, our method shows consistent improvements on the following classes in the target domain: "wall" (+0.4%), "fence" (+0.4%), "bicycle" (+0.7%), "vegetation" (+1.2%), "car" (+1.0%) and "motorbike" (+0.7%). Fig. 2 shows the results of the qualitative comparison of our method with DADA[12] and CTRL[13]. We also show the estimated results of depth and density in the figure. From the results of density estimation, the trace of segmentation can be seen.

### 4.3.2 SYNTHIA → Cityscapes (7 classes) and SYNTHIA → Mapillary (7 classes)

Table 2 shows the semantic segmentation results in SYNTHIA → Cityscapes and SYNTHIA → Mapillary benchmarks, which are trained and evaluated on their common 7 classes. We also train and evaluate our model on the $320 \times 640$ resolution, so as to make a fair comparison with the reference low-resolution models. In the low-resolution model in SYNTHIA → Mapillary benchmark, CTRL is superior to other methods, and our model ranks second with mIoU loss lower than CTRL by -2.2%. Besides, our proposed method is superior to previous works. Compared to CTRL, our mIoU gains are +2.3% and +1.2% for low-resolution and full-resolution of SYNTHIA → Cityscapes, and +1.2% for full-resolution of SYNTHIA → Mapillary. Fig. 3 shows the results of the qualitative comparison of our method with DADA[12] and CTRL[13] on these two protocols at full-resolution.

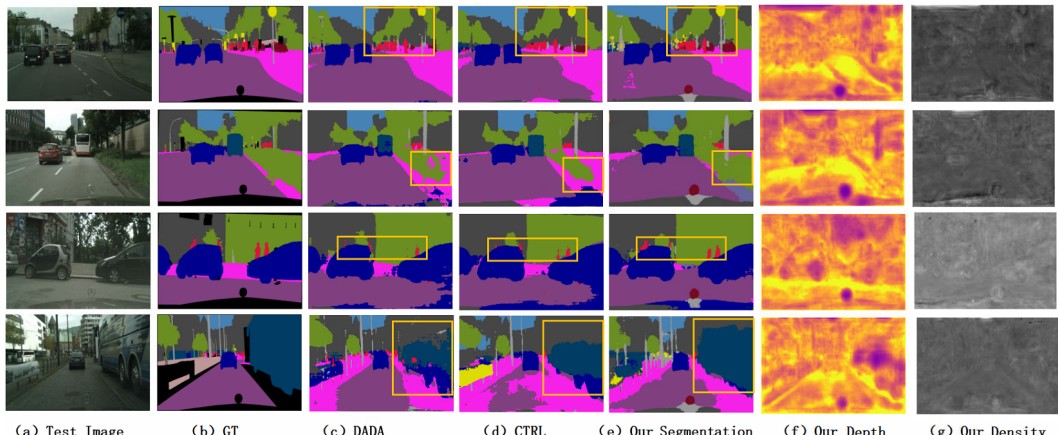

| (a) Test Image | (b) GT | (c) DADA | (d) CTRL | (e) Our Segmentation | (f) Our Depth | (g) Our Density |

Figure 2: Qualitative results on SYNTHIA → Cityscapes (16 classes). (a)Test images from Cityscapes validation set; (b) ground truth segmentation; (c) DADA[12] segmentation; (d) CTRL[13] segmentation; (e) our segmentation; (f) our depth; (g) our density. Compared to theirs, our method demonstrates notable improvements on "light and sign", "vegetation", "person" and "bus" classes as highlighted using the yellow boxes.

## 4.4 Ablation Studies

In order to verify the effectiveness of our proposed components, we trained four different models for comparison, denoted as M1,...M4. An ablation study is shown in Table 3. "SegPre" represents segmentation prediction branch, "DepRes" represents depth regression branch, "DenEst" represents density estimation branch, "SelfTra" means the self-training scheme in our paper without using spatial prior pseudo-labels refinement algorithm, "SpaPri" means the spatial prior pseudo-labels refinement algorithm. All the models are trained and evaluated on SYNTHIA → Cityscapes (16 classes).

M1 is backbone network with segmentation predict branch and depth regression branch. Unlike DADA, we didn't fuse features for adversarial training in M1. The result shows that segmentation result (mIoU 41.7) is lower than that of DADA (mIoU 42.6). M2 adds the branch of density estimation, and this is our model. The segmentation result is 44.8 (mIoU), which is a satisfactory improvement. Note that, for fair comparison, if without self-training, CTRL result is 42.1(mIoU), which can be found in their ablation study table[13]. Apart from three sub-tasks, CTRL has an additional Cross-Task Relation Layer, but we choose a more concise MTL framework. Compared to DADA, M2 has a mIoU gain of +2.2%. Compared to CTRL without self-training, M2 has a mIoU gain of +2.7%. This demonstrates the effectiveness of the depth distribution density branch proposed by us. M3 adds our self-training scheme by commonly used pseudo-labels refinement based on confidence, and M4 adopts our proposed spatial prior refinement algorithm on the basis of M3. The result of M3 demonstrates our model can obtain consistent improvements by using self-supervised training. In addition, compared to M3, M4 has a mIoU gain of +0.6%, which shows the effectiveness of our proposed algorithm to obtain higher quality pseudo-labels.

Table 3: Ablation study of different components of our method in Section 4.4

Table 4: Additional analysis in Section 4.5

| Model | SegPre | DepRes | DenEst | SelfTra | SpaPri | mIoU(%)↑ |
|-------|--------|--------|--------|---------|--------|----------|
| M1 | √ | √ | | | | 41.7 |
| M2 | √ | √ | √ | | | 44.8 |
| M3 | √ | √ | √ | √ | | 47.6 |
| M4 | √ | √ | √ | √ | √ | **48.2** |

| Situation | mIoU(%)↑ |
|-----------|----------|
| S1 | 44.1 |
| S2 | 43.4 |
| S3 | 37.8 |
| S4 | 43.7 |
| S5 | **44.8** |

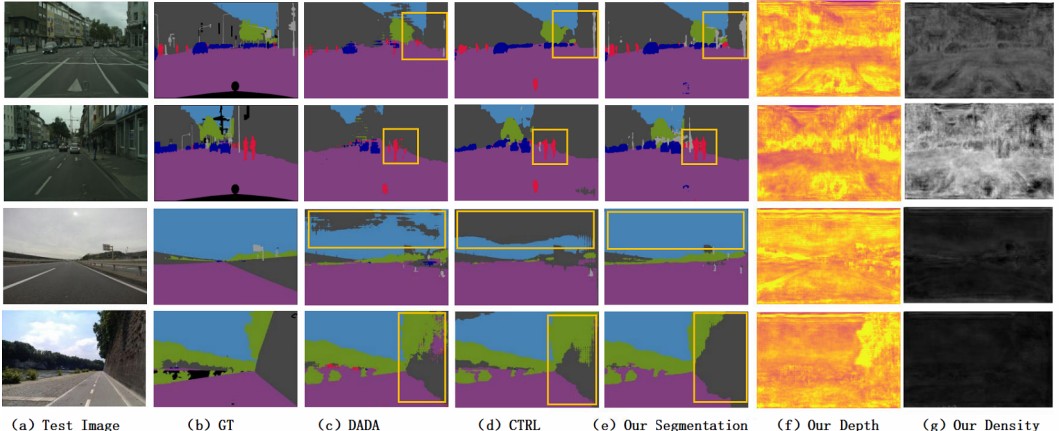

| (a) Test Image | (b) GT | (c) DADA | (d) CTRL | (e) Our Segmentation | (f) Our Depth | (g) Our Density |

Figure 3: Qualitative results on: SYNTHIA → Cityscapes (7 classes) (upper two rows) and SYNTHIA → Mapillary (7 classes) (lower two rows). (a) Test images from Cityscapes and Mapillary validation sets; (b) ground truth segmentation; (c) DADA[12] segmentation; (d) CTRL[13] segmentation; (e) our segmentation; (f) our depth; (g) our density. Compared to theirs, our method demonstrates notable improvements on "object", "human", "sky" and "construction" classes as highlighted using the yellow boxes.

## 4.5 Additional Experimental Analysis

**Branch balance loss for density estimation.** The novelty of this loss lies in the density map constructed for regression, which is calculated online during training. We propose the branch balance loss for multi-task learning, which is different from the SOTA methods. When the branch balance loss is removed, density regression on the source domain will use the pre-constructed density map as the ground truth for supervised training, and density regression on the target domain will not exist. The mIoU result of this model is 44.1 (S1 in Table 4). Note that S5 is the result of our model, and all results in Table 4 are without the self-training scheme on SYNTHIA → Cityscapes (16 classes).

**Ground truth depth for density map construction during source domain training.** For comparison, we use the predicted depth to construct the density map in the source domain training. The result of mIoU is 43.4 (S2 in Table 4), which is lower than the method we proposed. It is known that the density map is closely related to the segmentation map and the depth map. In our source domain training, we eliminate the influence of depth estimation by using its ground truth, but transfer the influence to segmentation by using the predicted segmentation results. This change makes the density map closely related to semantic segmentation, and thus improving the segmentation prediction performance.

**Features fused from density and semantic segmentation.** Our feature fusion strategy is not only different from DADA, which fuses depth and segmentation results, but also different from CTRL, which concatenates the features of three branches together. For comparison, we trained a model to fuse all results of our three branches together. The result of mIoU is 37.8 (S3 in Table 4). We believe that depth density with segmentation may enhance the antagonism of segmentation, while the depth information added to the fused feature may interfere with and weaken the main task segmentation, which is why the performance of S3 drops so much. In addition, we trained another model with the same feature fusion as DADA in our framework . The result of mIoU is 43.7 (S4 in Table 4). Therefore, our fusion strategy (S5 in Table 4) can highlight the effect of the main task.

**t-SNE comparison of features learned by two different joint spaces.** In Fig. 4, we use t-SNE [44] to visualize features in target domain through (a) joint space of depth and segmentation, and (b) joint space of density and segmentation. The two models are trained and evaluated following the UDA protocol SYNTHIA → Cityscapes (16 classes). For comparison, some classes are circled. The t-SNE results show that our density+segmentation joint space is better than depth+segmentation joint space, and it has a high degree of aggregation and a better class separation in the target domain.

**The influence between density and depth estimation.** To verify whether density can promote depth, we use M1 model and our M2 model in Table 3 to generate the depth results of the test images of

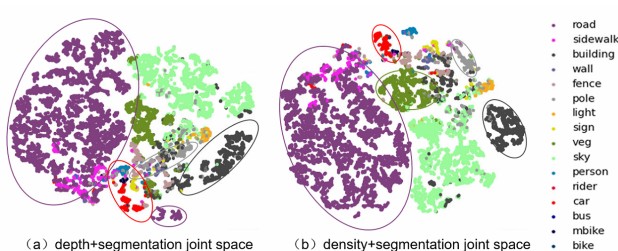

(a) depth+segmentation joint space     (b) density+segmentation joint space

road
sidewalk
building
wall
fence
pole
light
sign
veg
sky
person
rider
car
bus
mbike
bike

Figure 4: t-SNE comparison of features learned by two different joint spaces.

|  | M1 | M2 |
| --- | --- | --- |
| $|Rel|\downarrow$ | 0.7 | **0.5** |
| $Rel^2\downarrow$ | 13.7 | **9.0** |
| $RMS\downarrow$ | 20.7 | **18.3** |
| $LRMS\downarrow$ | 0.9 | **0.7** |
| $\delta_1\uparrow$ | 0.21 | **0.26** |
| $\delta_2\uparrow$ | 0.40 | **0.48** |
| $\delta_3\uparrow$ | 0.56 | **0.66** |

Table 5: Depth comparisons between M1 and M2 model in Table 3.

Cityscapes dataset. The depth results are shown in Table 5. As can be seen from the table, density estimation can improve depth results. It can be done from the following two aspects: First, the density and depth are closely related. In the process of building a bridge between two domains through depth density, depth prediction is naturally promoted. Secondly, in our training, our branch balance strategy closely links depth regression with density estimation. Therefore, the density helps to improve the depth.

**Sensitivity analysis of hyper parameters.** Because we use multi-task learning, the parameters of each task are selected as follows: the main task segmentation is a large weight, and we set it to 1.0. The two auxiliary tasks, density and depth, are small weights. Considering that density is used to fuse with segmentation, which means that density is more important than depth, we set the weight of density to $10^{-2}$, and the weight of depth to $0.5 \times 10^{-2}$. We also designed the experiments to increase the hyper parameters of two auxiliary tasks to the level of $10^{-1}$, or decrease to the level of $10^{-3}$, and the final results will be worse. However, the change of parameters at the level of $10^{-2}$ has little effect on segmentation.

## 4.6 Limitations and Discussions

**The training cost.** Our method is based on multi-task learning, and it will cause a relatively high cost, which is also inevitable in DADA[12] and CTRL[13] methods. DADA has two sub-tasks, while CTRL and our method have three sub-tasks. Compared to CTRL and ours, the training time of DADA is relatively short, and the memory utilization rate is low. The first stage of our training is $60K$ for almost 90 hours, and self-training for 10 hours. The GPU memory usage is about 10GB. However, by adding the branch of depth density estimation, our performance is improved.

**The results of depth estimation.** Depth estimation is an auxiliary task in our framework, and it is used with density to promote the main task. We use the same branch network as the previous methods[12, 13] for depth regression. Like theirs, we find that our depth regression results cannot be compared with the works focused on depth estimation. This is another limitation of our work. However, depth density is closely related to depth, so in our method, a more accurate depth and its distribution density should be able to further improve the performance of semantic segmentation. We plan to redesign the depth and density branch network in the MTL framework in our further work.

## 5 Conclusion

We use depth distribution to build a bridge between synthetic datasets and real-world datasets for semantic segmentation in UDA context. We predict three tasks using the standard MTL framework, and the two auxiliary tasks are designed to improve the performance of semantic segmentation. By using our method, a new SOTA segmentation performance is achieved.

**Acknowledgments.** This work is funded by the National Natural Science Foundation of China (No.41771427). We thank the anonymous reviewers for the valuable feedback and time spent.

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
