# 1 Supplementary Material

## 1.1 The analysis of the main potential technical contributions.

(1) We suggest using depth density to bridge the domain gap. Some existing methods, such as SPIGAN and DADA, use depth information in a plain way. CTRL divides the depth information into discrete depth levels, and it also lacks a detailed and continuous expression of the depth distribution. We use the depth density calculated by Gaussian mixture models (GMMs) to bridge these the two domains, which can provide a more detailed quantitative description. Our method, SPIGAN, DADA, CTRL, all have the problem that the depth values in the two domains cannot be strictly and completely consistent in the method with depth information as the domain bridge. In our method, GMMs are built for all images on the source domain instead of highlighting one of them. Therefore, the depth distribution reflects the probability that image pixels appearing in different classes in the whole domain. In addition, adversarial training is also a game between true and false probability, which weakens the influence of absolute depth to some extent. Therefore, the camera settings of the source domain and the target domain may be different, which will not have much impact on us.

(2) We propose the branch balance loss for density estimation. Its novelty lies in constructing density maps for regression from two aspects. Firstly, most of the existing hard parameter sharing methods for multi-task learning use independent loss function to constrain sub-tasks. However, our three sub-tasks can be directly connected by the density map we proposed. This breaks the existing mode of independent training of sub-tasks in multi-task learning, and can only be indirectly constrained by sharing the network layers (backbone). Secondly, in the adversarial training, the multi-task learning on the target domain is unsupervised, so the sub-tasks in the existing methods cannot be constrained on the target domain. Our method for constructing the density map can make the three branches interact, restrict and keep a balance during training, especially on the target domain.

(3) For spatial prior pseudo-labels refinement, we observe that most of the pixels in each class are aggregated and adjacent to each other. For large and obvious objects, such as the sky, land, road and bus, their pixels are concentrated in large area, while those of rare class are concentrated in small area. By using the different thresholds to judge the aggregation degree of pixels, the selection of pseudo-labels is strengthened. Moreover, our self-training strategy is heuristic, but instead of relaxing the constraints, we are stricter in the choice of pseudo-labels, as shown in Figure 1. We show the pseudo-labels refinement results by (d) only confidence and (e) our proposed algorithm based on (c) the results of M2 model. It can be seen that our algorithm is more strict than confidence refinement.

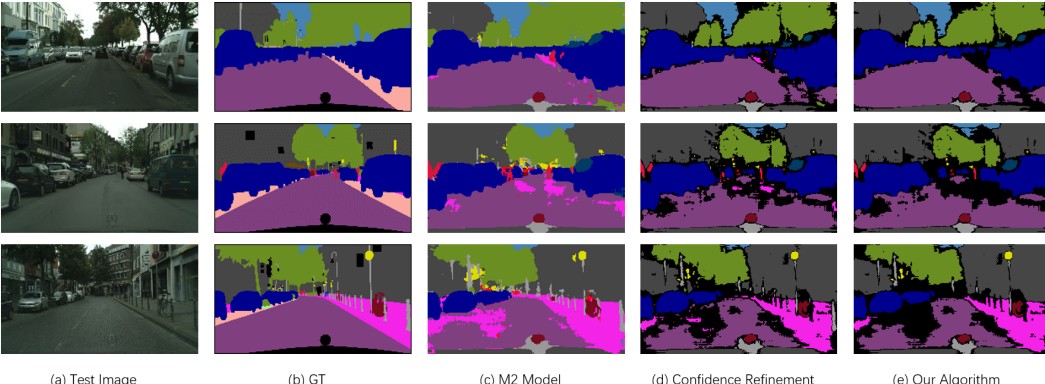

Figure 1: Qualitative results on SYNTHIA → Cityscapes(16 classes). (a) Test image from Cityscapes validation set; (b) ground truth annotations; (c) the results of our M2 model without self-training; (d) the results of pseudo-labels refinement only using confidence for the results of the M2 model; (e) the results of our spatial prior pseudo-labels refinement algorithm for the results of the M2 model; The black areas in (d) and (e) are filtered out pseudo-labels, and they will not be used for the next self-training.