# OpenReview forum: "Unsupervised Domain Adaptation for Semantic Segmentation using Depth Distribution"
_NeurIPS.cc/2022/Conference — NeurIPS 2022 Accept_

### Official Review · Reviewer_sfyu · 2022-07-10

**Rating:** 5
**Confidence:** 5
**Soundness:** 3 good
**Presentation:** 3 good
**Contribution:** 3 good

**Summary:**

This paper addresses the task of unsupervised domain adaptation in semantic segmentation. Following the same line as previous works [11,12,13], the paper leverages depth supervision in source domain as the privileged information to improve segmentation in target domain. The paper presents an intuitive idea based on the observation that: pixels of a particular semantic class often have similar positions in the image space and have similar depth ranges across source and target. Based on which, the paper proposes a task of estimating depth distribution density 2D maps, which are then used to guide the final output-level adversarial adaptation. Reported results are better than previous SOTAs.

** Technical details **
The network architecture is a standard MTL with three heads: one for the main task of segmentation, one for monocular depth and one for estimation the ``depth distribution density maps'' (see below for more details); the three branches are attached to the same Resnet101 backbone. The two branches segmentation and depth are supervised in source with cross-entropy and berHu losses as done in [12]. Depth distribution density supervision is gotten from segmentation prediction $\hat{P}$ and either ground-truth depth $Z_s$ in source or estimated depth $\hat{Z}_t$ in target.

The depth distribution of each semantic class is modeled as a Gaussian mixture model (GMM) in the image space; GMM parameters are estimated using source data. The number of components is fixed as 5. For each source image, using the depth ground-truth $Z_s$ and segmentation prediction $\hat{P}_s$, one can produce a depth distribution density map $D_s\in\mathbb{R}^{H\times W}$, i.e. at the position $(x,y)$, the value $D_s(x,y)$ is determined as the depth distribution density at $(x,y)$ of class $\hat{P}_s(x,y)$. For each target image, depth distribution density map $D_t$ is estimated using depth prediction $\hat{Z}_t$ and segmentation prediction  $\hat{P}_t$.

For adaptation, this work applies adversarial training using an additional discriminator [19,12]. Input to the discriminator is a fusion of segmentation prediction $\hat{P}$ and estimated depth density map $D$. To further boost performance with self-training, a heuristic pseudo-labelling procedure is used.

**Questions:**

Questions:
- Why berhu loss is used for supervising the density branch? What is the reaso?

Some analyses that could improve this work:
  - Sensitivity analysis of hyper parameters, especially the distribution density one.

  - Although depth and density estimations are auxiliary tasks, it would be interesting to quantify (as done in [13]) or at least visualize
  some depth and density results. Does density improve depth?

  - It's not yet been a standard for UDA in segmentation due to high computational cost but it would be highly
  appreciated if the authors can provide mean and std performance with different runs and seeds.

**Limitations:**

The conclusion discusses a general limitation on depth quality, same as similar works.
No discussion on potential negative societal impacts was given.

**Strengths And Weaknesses:**

** Strengths **
- The idea is intuitive and it seems easy to implement the density branch given the availability of existing codebase.
- Paper presentation is quite clear and easy to understand. However writing could be improved here and there, e.g. L144-146 I don't quite get the meaning.

** Weaknesses **
- This work lacks quantitative and qualitative analysis of the new branch for density distribution estimation. It's hard for me to imagine what are produced by the density branch. What advantages brought by the joint space of segmentation and density ( as compared to depth + seg joint space)? Some qualitative density results as well as t-SNE/UMAP visualization of joint-space features are needed.
- The self-training strategy is highly heuristic. Seemingly this strategy help recruit more rare-class pseudo-labels by relaxing the constraints for those (?) I don't understand why the values 50 and 4950 were chosen.

=========== Post-rebuttal:
I thank the authors for actively answering my questions during the rebuttal period. Most of my concerns were addressed. I'm happy to increase my score.
If this paper is accepted, I would like to see in the final version more discussions and qualitative results as promised by the authors, especially on the pseudo-labelling refinement.

---

> ### Author Response · Authors · 2022-08-02
> **Please refer to the comments.**
>
> [Q1] Regarding the  L144-146.
>
> [A1]  We modeified the whole paragraph, removed the unclear sentences. "We propose a branch balance loss for the depth distribution density regression, and its novelty is the different calculation ways of the constructed density map D. For source domian training, instead of using the precalculated density maps of the source samples, we use... For target domain training, we use ..."
>
> [Q2] Regarding the new branch for density distribution estimation.
>
> [A2] According to your suggestion, we designed an experiment of depth fused with segmentation under our framework, and the mIoU result is 43.65, while the mIoU result of our fusion strategy is 44.8. The experiment shows that the joint space of density + segmentation is better than the joint space of depth + segmentation. In Figure 2 and Figure 3, we also added the results of density and depth results.  The new figures 2 and 3 have been uploaded on our source code website. In addition, we use t-SNE to visualize features in Cityscapes database (target domain) through density+segmentation joint-space and depth+segmentation joint-space. The t-SNE results show that our density+segmentation joint space is better than depth+segmentation joint space, and it has a high degree of aggregation and a better class separation in the target domain. For comparison, some classes are circled. All the visualization results have been updated on the website and will be added to the final paper.
>
> [Q3] Regarding the self-training strategy.
>
> [A3] Our self-training strategy is heuristic, but we think that instead of relaxing the constraints, we are stricter in the choice of pseudo-labels. We observe that most of the pixels in each class are aggregated and adjacent to each other. For large and obvious objects, such as the sky, land, road and bus, their pixels are clustered on a large area, while those of rare-class are clustered on a small area. By using the different thresholds to judge the aggregation degree of pixels, the selection of pseudo-labels is strengthened. Default values are set as 50 and 4950, so the threshold range is 50 to 5000.  As the aggregation scale of different categories varies greatly, we set a threshold range of 1：100 according to experience. Of course, this can be further adjusted according to the actual size of classes in different datasets.
>
> [Q4] Why berhu loss is used for supervising the density branch?
>
> [A4] Density map and depth map are similar in numerical expression. This inspires us to use berhu loss for density regression. Actually, we've tried L1 loss and Smoothed L1 loss, but the effects are not as good as berhu loss.
>
> [Q5] Regarding the sensitivity analysis of hyper parameters.
>
> [A5] As we use multi-task learning, the parameters of each task are selected as follows: the main task segmentation is a larger weight, and we set it to 1.0. The two auxiliary tasks, density and depth, are small weights. Considering that density is used to fuse with segmentation, which means density is more important than depth, we set density weight to 10-2, and the depth weight to 0.5*10-2. We designed the experiments to increase the parameters (depth and density) to the level of 10-1, or decrease to the level of 10-3, and the final result will be even worse. However, the change of parameters at the level of 10-2 has little influence on segmentation. We will include the above analysis in the final paper.
>
> [Q6] Regarding the depth and density.
>
> [A6] To verify whether density can promote depth, we use M1 model (depth+segmentation) and M2 model (depth+density+segmentation）in Table 3 to generate the depth results of the test images of Cityscapes dataset, the results for depth are as follows:
>
>          abs_rel|sq_rel| rmse | rmse_log |  a1 |  a2 | a3 |
>
> M1     0.683   &  13.658  &  20.741  &   0.905   &   0.211  &   0.403  &   0.563
>
> M2     0.536   &   8.995   &  18.281  &   0.673   &   0.256  &   0.480  &   0.663
>
> It shows density can improve depth. In addition, we update the visualization results of depth and density in Figure 2 and 3, which have been uploaded on the source code website and will be put in the final paper.
>
> [Q7] Mean and std performance with different runs and seeds.
>
> [A7] Honestly, due to high computational cost, we didn't test the mean and std performance with different runs and seeds. We admit that similar work [13] has done an admirable work in this regard.
>
> [Q8] Discussion on potential negative societal impacts.
>
> [A8] We will add more limitation discussion, such as training cost, memory usage, etc., in the final paper according to the comments of reviewers.
>
> Thank you for your careful review and giving the opportunity to improve the final paper!

---

> > ### Comment · Reviewer_sfyu · 2022-08-08
> > **Discussion on depth and density results**
> >
> > Dear authors, thanks for your efforts addressing some of my concerns. There still remains a few as follows:
> >
> > 1) May the authors elaborate more on why density estimation helps improve depth estimation?
> >
> > 2) I thank the reviewers for providing the qualitative results of depth and density in the repo; but it's hard for me to interpret the results only looking at those figures. First, how to understand the heatmap colors? Second, could the authors point out which density areas of the given density examples are particularly favorable for depth, for segmentation or for UDA?
> >
> > 3) Self-training. I checked the self-training code here: https://github.com/depdis/Depth_Distribution/blob/main/depth_distribution/main/domain_adaptation/selftrain_UDA.py . It seems that validation set is used to find the best model at each self-training stage. Does that imply certain supervision leak from the target domain? In my opinion, having this kind of model selection done once at the very end could be acceptable, but in the course of self-training one should not use the validation set. Taking the last model seems more appropriate for self-training. May the authors comment on this concern?

---

> > > ### Author Response · Authors · 2022-08-08
> > > **Please refer to the comments.**
> > >
> > > It's our pleasure to answer your questions, and we cherish the opportunity to communicate with you!
> > >
> > > [Q1] Regarding the density estimation helps improve depth estimation.
> > >
> > > [A1] We believe that the density estimation is helpful to improve depth estimation, which can be done from the following two aspects: First, the density is closely related to depth and segmentation, and density is "the density of depth". In the process of building a bridge between two domains through depth density, the prediction of depth will naturally be promoted. Secondly, in our training process, we use the online predicted depth to construct depth density on the source domain and target domain, which is the advantage of our designed branch balance loss. (Line 146-150) Especially on the source domain, we use the predicted depth instead of the ground truth depth to construct the density map. This training process closely links the depth prediction with density prediction. Therefore, density estimation is helpful to improve depth estimation.
> > >
> > > [Q2] Regarding the visualization and favoras of the density map.
> > >
> > > [A2] Thanks for your questions, we will give a detailed explanation here. To visualize the density map, we use the following codes to visualize each pixel.
> > >
> > > opencv_data = (
> > >
> > > (0.000, (1.00, 0.00, 0.00)), # red
> > >
> > > (0.400, (1.00, 1.00, 0.00)), # yellow
> > >
> > > (0.600, (0.00, 1.00, 0.00)), # green
> > >
> > > (0.800, (0.00, 0.00, 1.00)), # blue
> > >
> > > (1.000, (0.60, 0.00, 1.00)) # purple
> > >
> > > )
> > >
> > > Therefore, the lower the value, the red color will be displayed, while the higher the value, the purple color will be displayed. Please note that most density probability values calculated by pixels are ranging from 10-4 to 10-5. For the convenience of display, we normalize them between 0 to 1.
> > >
> > > Your second question is really very good. Each pixel in the density map describes the probability of seeing the depth value with this semantic class at this pixel location under the learned GMM. We can't say it is favorable for depth or for segmentation, but if it is visualized as a grayscale image, you can see the trace of segmentation. We have uploaded the grayscale images of Figure 2 and Figure 3 on the website. However, specifically speaking, we can say it favorable for the adversarial training. The depth value of each pixel has a different probability in its own class. In the process of reconstructing the density map,  we can't say that the value with high probability is right, and the value with low probability is wrong, because only the real distribution of probability value can be valuable for adversarial training.
> > >
> > > [Q3] Regarding the self-training.
> > >
> > > [A3] There may be some misunderstanding about our self-training process. In the "selftrain_UDA.py", we have indicated that "#eval， can be annotated", and the best model is only for the final evaluation after two round of self-training. Although we can test the best model in each round, we don't take it as our next training. We adopt the similar "Iterative Self Learning process" with [13], but the difference is our model needs two rounds of 30K self-training(Lines 202,203,223), and uses the spatial prior pseudo-labels refinement algorithm. So the last model of the first round is used for self-training in the second round. In our paper, this strategy and the parameters are the same for three standard UDA evaluation protocols: SYNTHIA → Cityscapes (16 classes), SYNTHIA → Cityscapes (7 classes), and SYNTHIA → Mapillary (7 classes). You can also annotate two lines "model.eval()" and "best_miou, best model = evaluate_domian_adaptation()" in "selftrain_UDA.py" to verify our model.
> > >
> > > Thank you for your questions. If there is something unclear, we are happy to continue to answer!

---

> > > > ### Comment · Reviewer_sfyu · 2022-08-09
> > > > **Thanks for the answers**
> > > >
> > > > Thanks the authors for the prompt answers that clear my concerns.

---

> > > > > ### Author Response · Authors · 2022-08-09
> > > > > **Thank you for your careful review.**
> > > > >
> > > > > Thank you for your careful review. We will put the grayscale image of density map and the above analysis in the final paper!

---

### Official Review · Reviewer_pVBs · 2022-07-10

**Rating:** 4
**Confidence:** 4
**Soundness:** 3 good
**Presentation:** 3 good
**Contribution:** 1 poor

**Summary:**

For the UDA task, the authors propose to utilize the probability density of depth distribution to bridge the domain gap for semantic segmentation.
The proposed method conduct experiments on the SYNTHIA-to-Cityscapes and SYNTHIA-to-Mapillary setting.

**Questions:**

What is the training cost when involving the additional branches?

Is the model sensitive to the weight values in Line166? How do you pick those parameters?

After reading the whole paper, It is still very hard to understand the density of depth distribution. I highly recommend adding a visualization on the density map, and it might be helpful for readers to follow the overall idea.

**Limitations:**

Using additional branches for training introduced more time and resources to get a model.


**Strengths And Weaknesses:**

(+) The code is provided.
(+) Writing and presentation are good.
(+) The branch balance strategy and Pseudo-labels refinement are interesting.
(-) A lot of recent UDA approaches, e.g. [R1-R4] are missing in this paper. The proposed	method can not perform as well as many CVPR2021/CVPR2022/ICCV2021 works.
(-) The most common case under the UDA setting -- GTA5->Cityscapes is missing. I suspect the performance of the proposed method in this setting.

[R1] Zhang, Pan, et al. "Prototypical pseudo label denoising and target structure learning for domain adaptive semantic segmentation." Proceedings of the IEEE/CVF conference on computer vision and pattern recognition. 2021.

[R2] Huo, Xinyue, et al. "Domain-Agnostic Prior for Transfer Semantic Segmentation." Proceedings of the IEEE/CVF Conference on Computer Vision and Pattern Recognition. 2022.

[R3] Hoyer, Lukas, Dengxin Dai, and Luc Van Gool. "Daformer: Improving network architectures and training strategies for domain-adaptive semantic segmentation." Proceedings of the IEEE/CVF Conference on Computer Vision and Pattern Recognition. 2022.

[R4] Wang, Qin, et al. "Domain adaptive semantic segmentation with self-supervised depth estimation." Proceedings of the IEEE/CVF International Conference on Computer Vision. 2021.

---

### Official Review · Reviewer_WoDq · 2022-07-11

**Rating:** 6
**Confidence:** 4
**Soundness:** 3 good
**Presentation:** 3 good
**Contribution:** 3 good

**Summary:**

This work is about unsupervised domain adaptation for semantic segmentation and how to improve the domain transfer with depth information. In contrast to prior work, the authors investigate the distribution of depth for individual categories as a better bridge between domains than plain depth prediction. Specifically, a Gaussian Mixture Model (GMM) is fit to the source training data (x-y location and depth of each pixel, separately for each class), which is then used to compute a per-pixel density map, i.e., the probability of observing a depth value at a certain pixel location and with a certain semantic class. Then, a neural network is extended to not only predict semantics and depth, but also the depth distribution. For domain adaptation, an adversarial loss is computed on features that are a combination of the semantic and depth distribution branch. Finally, a spatial-aggregation prior to compute better pseudo labels for self-training is also proposed. Experiments on standard benchmarks (that include depth ground truth in the source domain) show state-of-the-art results and an ablation study investigates the impact of the individual contributions.

**Questions:**

- I'm wondering if the same idea of adding depth distribution as additional task would help plain semantic segmentation on the source domain, i.e., in a non domain adaptation setting.
- Line 142: What's a "channel-wise GMM"?
- Is there any explanation why the performance drops so much for S3 in Table 4? I was actually wondering what the results look like if you did the same feature fusion as DADA.

**Limitations:**

One limitation has been discussed in the conclusion (sub-optimal architecture design for depth prediction branch). Potential societal impacts were not discussed.

**Strengths And Weaknesses:**

### Strengths

- Well written, except a few grammatical issues and typos (please get those checked)
- The structure of the paper is well done, illustrations help understand the method
- Ablation studies are mostly good
- Main results show a good improvement over state-of-the-art
- Source code is made available

### Weaknesses

- The motivation about class-wise depth distribution in lines 41ff seems to be based heavily on typical driving scenes. It would be better to also think about other types of scenes (e.g., indoor) and see if the same argument holds.
- Throughout the paper, it was hard to imagine what "spatial aggregation priors" could mean until reading the actual section about it. Similarly in line 100, "the spatial aggregation scale of pixels" is unclear to me.
- The depth distribution is advertised as domain bridge, but doesn't this modality depend on the camera settings, which may be different in source and target domains? For instance, if the focal length is different (i.e., more or less zoom), then objects further away would be represented with more pixels and thus likely change the depth distribution.
- In nearly all equations, it would be better to end it with a comma (",") instead of having the comma in the beginning of the next line.
- For equation 5, I think it would be good to add an intuitive explanation that every pixel in D describes the probability of seeing the depth value with this semantic class at this pixel location under the learned GMM.
- Lines 155f: The argument about "balance" of branches is unclear to me, and also why the loss is named that way.
- Section 3.5 needs to be re-written. First, it was not clear to me that self-training would be part of the proposed solution. Second, in line 189, it's not clear initially what the thresholds are for and how the pseudo labels are "refined" with them. Finally, the spatial prior refinement algorithm needs a more intuitive explanation and potentially a figure for easier understanding.
- I do not understand the point the first paragraph in Sec. 4.5 (lines 267ff) tries to make.

---

> ### Author Response · Authors · 2022-08-02
> **Please refer to the comments.**
>
> [Q1] The motivation about class-wise depth distribution on other types of scenes (e.g., indoor).
>
> [A1] Thanks for your advice. Indeed, we only consider the class-wise depth distribution on typical driving scenes and will make a rigorous expression in the final paper. We will find the relevant indoor scenes datasets to verify the validity of the same motivation.
>
> [Q2] Regarding the "spatial aggregation priors" and "the spatial aggregation scale of pixels".
>
> [A2] They have the same meaning.  We will change another explicit expression in the final paper and explain it when it first appears. Since most pixels in the same class are clustered together in the image space, we calculate the total proportion of class-wise pixels in the source images, and use this information as a prior of pseudo-label refinement.
>
> [Q3] Regarding the camera settings.
>
> [A3] Our method, SPIGAN, DADA and CTRL all have the problem that the depth values in the two domains can not be strictly and completely consistent in the method of using depth information as the domain bridge. In our method, GMMs are built for all images on the source domain instead of highlighting one of them. Therefore, the depth distribution is a probability of image pixels appearing in different classes reflected in the whole domain. In addition, we normalize the images on the target domain to the size of the image on the source domain to minimize the difference between domains. In addition, we believe that the adversarial training is also a game between true and false probability, which also weakens the influence of absolute depth to some extent.
>
> [Q4] Regarding the comma and explanation of equation 5.
>
> [A4] We will revise and proofread this paper according to your suggestion and present a satisfactory final version.
>
> [Q5] Regarding the name of balance loss.
>
> [A5] The novelty of branch balance training lies in constructing density maps for regression. In the adversarial training, the multi-task learning on the target domain is performed unsupervised, so the subtasks in the existing methods are not constrained during the training on the target domain. The proposed method of constructing density maps during training can allow the three branches to influence and restrict each other, and keep their balance during the training, especially for the target domain (lines 157-160). We are glad to get you suggestion.
>
> [Q6] Regarding the Section 3.5.
>
> [A6] Self-training is a part of the method we proposed. After multi-task learning and adversarial training, self-training can further improve the segmentation effect. According to your suggestion, we will rewrite this part, increase the logic of its expression, and present a satisfactory final version.
>
> [Q7] Regarding the first paragraph in Sec. 4.5 (lines 267ff).
>
> [A7] We designed the experiment to verify the effect of branch balance loss. The novelty of this loss lies in the density map constructed for regression, which is calculated online during training. If the branch balance loss is removed，density regression on the source domain will use the pre-constructed density map, and density regression on the target domain will not exist.
>
> [Q8] Regarding the idea for semantic segmentation on the source domain.
>
> [A8] To be honest, we didn't test it on the source domain in the non-domain adaptation setup, but we think it's worth a try.
>
> [Q9] Regarding the channel-wise GMM.
>
> [A9] We will change it to "class-wise GMM" in the final paper.
>
> [Q10] Regarding the S3 in Table 4 and the same feature fusion as DADA.
>
> [A10] S3 in Table 4 is the fusion of three feature maps, including segmentation, depth and density.  In our method, Depth density is based on depth and segmentation. We believe that depth density with segmentation may enhance the antagonism of segmentation, while the depth information added to the feature may interfere with and weaken the main task segmentation. We think this is why the performance drops so much for S3. In CTRL, they concatenate segmenation, refiend segmenation and depth together, which also highlights the main task segmentation.  According to your suggestion, we added an experiment, and used the same feature fusion as DADA, that is, depth and segmentation are fused together. The mIoU result is 43.65 and the trained model has been uploaded on the website, while our M2 model is 44.8.
>
> Thank you for your approval of our work, we will carefully upgraded the final paper.

---

> > ### Comment · Reviewer_WoDq · 2022-08-09
> > **Appreciation of author feedback**
> >
> > I appreciate the author's feedback, which answers many of my questions and concerns. Overall, I'm still leaning towards acceptance of the paper.

---

### Official Review · Reviewer_jvKa · 2022-07-12

**Rating:** 5
**Confidence:** 4
**Soundness:** 4 excellent
**Presentation:** 3 good
**Contribution:** 2 fair

**Summary:**

This paper proposes a novel segmentation framework that could be aware of depth distribution density and thus improve the segmentation performance using unsupervised domain adaptation. The proposed network is based on the multi-task learning architecture that enables three different tasks: segmentation, depth regression, and depth density distribution estimation. At the end of the architecture, the adversarial network is appended, and thus the segmentation performance is significantly improved. The main contributions of this paper are (1) the suggestion of the probability density of depth distribution to bridge the gap in the unsupervised domain adaptation, (2) a novel idea of branch balance training that enables improving the segmentation performance, and (3) pseudo-labels refinement algorithm that utilizes the aggregation priors of pixels in the different categories on the source images. The evaluation was performed using synthetic images as the source domain and real-world images as the target domain. The experiments demonstrate the effectiveness of the proposed method in the unsupervised domain adaptive semantic segmentation task.

- The proposed architecture mainly aims at segmenting target objects in the images from the target domain, which has not been annotated from the source domain's knowledge (especially depth distribution density), which includes the annotated labels of segmentation map and depth distribution.

- The architecture is based on the simple multi-task learning architecture but utilizes the additional adversarial training architecture to improve the segmentation performance.

- In multi-task learning, three tasks of segmentation, depth regression, and estimating the depth distribution density are included.
  - The segmentation task is the same as the general semantic segmentation task (supervised learning with annotated labels) using cross-entropy loss.
  - The depth regression is also supervised learning with the annotated labels, and the regression is performed using the berHu loss function.
  - From the segmentation map and the depth information, annotation for the density distribution can be generated. Then, the depth distribution density estimation is performed using the estimated density from the other two branches and the predicted density (branch balance training).

- The proposed method was evaluated using two datasets of Cityscapes and Mapillary as the target domain. In addition, the SYNTHIA dataset is utilized as the source domain. The images in the target domain are real-world images, whereas the images in the source domains are generated synthetic images.

- The experiments report the state-of-the-art performance of the proposed depth distribution-aware unsupervised domain adaptation network compared to other state-of-the-art deep learning models.


**Questions:**

* Questions & Discussions
0. The main questions and discussions are raised by the vague explanations or ignorance of the reviewer. Clear explanations and the discussions in the rebuttal/discussion phase would resolve the following questions. If the reviewer misunderstands the concepts or definitions, please fix the problems with clearer explanations.

1. The definition of the domain adaptation should be more discussed. In this paper, the authors addressed that the proposed framework could bridge the domain gap in UDA for semantic segmentation. The domain adaptation task mainly aims at bridging the domain gap such as style differences, geometric differences, or even statistical gaps between two domains. However, this paper has not discussed the main domain gap to be resolved. The authors repeatedly advert the utilization of depth information to bridge the domain gap. The clear explanations that the utilization of depth information can bridge the “what domain gap” in this paper should be discussed. The reviewer expects that the manuscript could be improved with clear illustrations of the problem definition.

2. The main technical contribution should be more discussed.

- In the first contribution (Line 61-62; Page 2), the authors address that the depth distribution is similar in the source and the target domain. The reviewer agrees that this conjecture is well working while regarding the large object and obvious objects like sky, land, road, etc. However, it is worried that this conjecture could be well appliable to the small objects like bikes and pedestrians, frequently appeared objects in one domain, and many objects from the same category (e.g., many cars are in one image). Depth distribution density is based on this hypothesis, the authors should illustrate the experimental or mathematical evidence for this hypothesis. The authors mentioned this in Line 104-106; Page 3, but the statistical analysis and evidence (table or graph) should be provided.

- The second contribution proposed by the authors is the branch balance training, which performs simple density regression using the existing loss function. It would be better to explain why the training method using a simple loss function can contribute to NeurIPS society. In addition, the novel idea beyond the simple equation (equation 6) should be discussed in the manuscript.

- The reviewer was impressed by the novel idea of the utilization of depth distribution density to improve the segmentation performance. Despite the simple architecture based on multi-task learning, the motivation and the proposed methodology could be a novelty. Therefore, the forest could be the contribution rather than the trees.

3. The training pipeline and the network architecture should be discussed.

- The training procedure should be clearer. The proposed architecture can be divided into two pipelines: (1) multi-task learning architecture and (2) adversarial network architecture. In the multi-task learning architecture, the images in the source domain are used to train the network with their annotations (segmentation map, depth map, and the calculated density map). In contrast, the images in the target domain are used to train the network with a segmentation map (?), depth map (?), and the density map of the source domain. Then, it is concluded that the training process using the target domain only optimizes the parameters in the density-predictive pipeline, is it? Furthermore, the adversarial network architecture inputs the fused density map from the source and the target domain both. Therefore, it is concluded that the training process using the source and the target domains are concurrently performed on both (1) and (2) architectures. Is it truly the process to train the network? Other domain adaptation methods train the network using the following in general:

  - (a) Transfer the style of the source domain to that of the target domain and then train the network (M) with the property of the target domain. Then, M predicts the segmentation output using the images in the target domain. [1]

  - (b) Train the network (M) using the source domain. Then, transfer the style of the target domain to that of the source domain, and M predicts segmentation maps using the transferred images. [2, 3]

  - (c) CNN architecture generates feature maps from the source and the target domains, and the CNN architecture aims at increasing the similarity between two feature maps [4].

- The reviewer is interested in whether the proposed domain adaptation method is more similar to (a), (b), (c), or other existing methods (please list the reference). Or, if the proposed methods are significantly novel, then the training pipeline should be clearly illustrated. The reviewer recommends authors illustrate the training pipeline in a visual way (figures) or mathematical formulations (or algorithm table).

- Effectiveness of the adversarial training. Related to the question above, the reviewer is curious about the effectiveness of the adversarial training. As the reviewer understood, the adversarial training utilizes the combined feature maps of the density map and the predicted segmentation map. The authors should clearly describe the goal of the adversarial training. In the Generative Adversarial Network (GAN), the generator and the discriminator are simultaneously optimized by the min-max game. In the proposed network, what advantage could be realized by deceiving the discriminator here? Does the author aim at designing that the generators (here segmentation and density pipelines) aim at generating similar feature maps using the source and the target domains? In this case, the bias (biased to segmentation map or biased to density map) could be discussed.

- Cost of the proposed framework. The reviewer is curious about the training time, prediction time, and memory resources. Inevitably, the multi-task learning-based network causes a heavy cost in prediction/training time and even in memory utilization. Quantitative analysis or discussion can significantly improve the quality of the manuscript. For the reviewer, since the proposed network includes multi-task learning- and adversarial learning-based architecture, the training and prediction cost can be extremely burdened compared to the conventional algorithms.

[1] Lee, Seunghun, et al. "ADAS: A Direct Adaptation Strategy for Multi-Target Domain Adaptive Semantic Segmentation." Proceedings of the IEEE/CVF Conference on Computer Vision and Pattern Recognition. 2022.

[2] Hou, Yunzhong, and Liang Zheng. "Source free domain adaptation with image translation." arXiv preprint arXiv:2008.07514 (2020).

[3] Lee, Kyungsu, Haeyun Lee, and Jae Youn Hwang. "Self-Mutating Network for Domain Adaptive Segmentation in Aerial Images." Proceedings of the IEEE/CVF International Conference on Computer Vision. 2021.

[4] Ganin, Yaroslav, and Victor Lempitsky. "Unsupervised domain adaptation by backpropagation." International conference on machine learning. PMLR, 2015.


**Ethics Review Area:**

["I don’t know"]

**Limitations:**

1. Major issues
- Related to the question illustrated in "Question" the followings should be improved. (1) The definition of the domain adaptation and the domain gap should be discussed. (2) Limited contribution and novelty. (3) Vague explanations for the training pipelines.

- Additional visual illustrations are required. Specifically, the predicted segmentation maps are well organized in the experiment parts. However, the outputs of the proposed network are segmentation, density, and depth maps. Therefore, the author should exhibit three maps to demonstrate that the proposed network is outstanding in the unsupervised domain adaptive semantic segmentation task.

- Mathematical notations and abbreviations should be strict.

  - The mathematical modeling should be invited for newly proposed terms with precise definitions. For instance, the authors proposed depth distribution density and should provide a clear definition of this term. Likewise, "density map" and "depth map" should be invited with clear mathematical definitions, including rank, dimension, size, range, and type (matrix, vector, set, scalar, etc.).

  - It is better to use the familiar term: "annotation/ ground truth/ semantic label/ ground truth segmentation map/ ground truth semantic label" instead of "ground truth segmentation annotation". In this case, P_hat can be the "predicted segmentation map" or "segmentation prediction".

  - In Line 135; Page 4, X_i should be re-checked. As the authors notated, X_i depends on i. However, the authors described X_i as a tuple (x, y, z) which is regardless of i. Therefore, the reviewer think X_i should be the list of tuples (x, y, z) where the associated pixel is classified as i-th class.

  - In Line 136; Page 4, j is firstly invited in Pi_{i,j} without any definition. Please clarify the definition of j.

  - In Line 140, Page 4, does N(~~~) indicate the flag (0 or 1) whether X_i is the gaussian distribution? In this case, if the input of the function N is only X_i, the author can omit the mean and the variance, even though N is adopted from the notation of the normal distribution "N~(0,1)".

  - GMM first appeared in Line 84; Page 2 without any definition. Please fix it with the gaussian mixture model in Line 46; Page 2.

  - Algorithm I is significantly far from the mathematical modeling and algorithm. Please simplify the table.

2. Minor issues (recommendations)
- Justify the hyper-parameter selections (Line 165-166; Page 5/ Line 179-190; Page 5/ Line 192-193; Page 6/ Line 198-203; Page 6). In addition, since the architecture is based on the multi-task learning architecture and the memory of GPU is 11GB, the training should be conducted in small batch size. The batch size and the normalization method (batch/group/instance/etc normalization) should be discussed in this case.

- (if available) The experiments should be improved. The reviewer already understands that the evaluation metric of "mean Intersection over Union (mIoU)" can qualitatively measure the object details. However, recent studies [1-3] proposed new evaluation metrics to quantitatively measure the object detail (especially the boundaries of the target objects). Otherwise, the visualization of the feature map could illustrate the novel feature extraction when processing object details. Alternatively, the authors could refer to the activation maps [4]. More experimental or mathematical evidence should be justified to clearly report the outstanding segmentation performance in terms of the improved boundary-oriented segmentation.

[1] Fernandez-Moral, Eduardo, et al. "A new metric for evaluating semantic segmentation: leveraging global and contour accuracy." 2018 IEEE intelligent vehicles symposium (iv). IEEE, 2018.

[2] Lee, Kyungsu, et al. "Boundary-oriented binary building segmentation model with two scheme learning for aerial images." IEEE Transactions on Geoscience and Remote Sensing 60 (2021): 1-17.

[3] Cheng, Bowen, et al. "Boundary IoU: Improving object-centric image segmentation evaluation." Proceedings of the IEEE/CVF Conference on Computer Vision and Pattern Recognition. 2021.

[4] Selvaraju, Ramprasaath R., et al. "Grad-cam: Visual explanations from deep networks via gradient-based localization." Proceedings of the IEEE international conference on computer vision. 2017.


3. Simple recommendations
- The reviewer recommends reviewing the grammar and typo errors to improve the quality of the manuscript.

- The reviewer recommends clarifying the meaning of the arrows in figure 1 (dashed line and other lines). Especially, L_bal uses the calculated density and the predicted density. What does "entropy" indicate? The inputs and outputs of each module are recommended to be clarified.

- Motivation for utilization of the berHu loss, ResNet101, ASPP, and AvgPooling.

**Strengths And Weaknesses:**

1. Strengths
- The reviewer agrees that the significance and importance of the task addressed in this paper, especially the depth information in each category, can improve the segmentation performance. In addition, unsupervised domain adaptive semantic segmentation with depth distribution density has rarely been studied to our best knowledge, such that the idea of the proposed task and network in this paper would be a novel contribution. Furthermore, the experimental result reports the state-of-the-art performance of the proposed network, and thus the proposed network can be expected as a novel framework.

- Additionally, the problem definition is clear. The authors were motivated to utilize the depth information and the depth density distribution while segmenting the target object. The idea of using depth distribution density is simple yet effective.

- The manuscript is well organized and well written.


2. Weakness
- More detailed descriptions are illustrated in the “Limitation” section. The main limitation suggested by the reviewer is the lack of detailed description. Please see below.

---

> ### Author Response · Authors · 2022-08-02
> **Please refer to the comments.**
>
> [Q1] Regarding the definition of the domain adaptation.
>
> [A1] We use depth information to bridge the domain gap on geometric differences, which is the same as SPIGAN[11], DADA[12] and CTRL[13]. In our related work, we use a section "Use of geometric information in semantic segmentation" for demonstration. We will give a clear explanation in the final paper.
>
> [Q2] Regarding the experimental evidence for similar depth distribution density.
>
> [A2] Before we put forward this hypothesis, we used GMMs to model on SYNTHIA and Cityscapes datasets. The results of GMM parameters on different classes are stored in "GMM parameter" file which has been uploaded to the source code website. It can be found that most classes have the similar GMM form. Because GMM of our method is 4-dimensional, and it is not easy to visualize, we will hyperlink the file to the final paper.
>
> [Q3] Regarding the contribution to NeurIPS society for branch balance training.
>
> [A3] The novelty of branch balance training lies in constructing density maps for regression. In our opinion,  it can contribute to NeurIPS society in two ways. Firstly, the existing multi-task learning with hard-parameter sharing way is to use independent loss function to constrain subtasks.  However, our three subtasks can be directly connected by our proposed density map (line 147-150). This breaks the existing mode of subtask independent training in multi-task learning, and can only be indirectly constrained by sharing the network layers (backbone). Secondly, in the adversarial training, the multi-task learning on the target domain is unsupervised, so the subtasks in the existing methods can not be constrained on the target domain. Our method for constructing the density map can allow the three branches to influence and restrict each other, and keep their balance during training, especially on the target domain (lines 157-160). We will include the above analysis in the final paper.
>
> [Q4] Regarding the contribution of utilizating of depth distribution density.
>
> [A4] Yes, some existing methods such as SPIGAN and DADA use depth information in a plain way. CTRL divides the depth information into discrete depth levels, and it also lacks a detailed and continuous expression of the depth distribution. We use the depth density calculated by GMMs to bridge the two domains, which can provide a more detailed quantitative description. Thank you for recognizing the potential contribution of our approach.
>
> [Q5] Regarding the training pipeline and the network architecture.
>
> [A5] There may be some misunderstanding about our training process.  Firstly, the calculation method used to construct the density map D is different in two domains (line 147-150), although the GMM of the source domain is used in these two domains (line 160-161). Therefore, it is necessary to calculate the reconstructed density map in the training process. Secondly, our method is different to (a),(b) and (c) mentioned by you, but similar to the existing methods DADA[12] and CTRL[13] mentioned in our paper.
>
> [Q6] Regarding the effectiveness of the adversarial training.
>
> [A6] It is also related to the above question. DADA uses segmentation + depth features. CTRL uses segmentation+refined segmentation+depth features. We use segmentation+density. According to your suggestion, we have added three experiments on different adversarial features, which are segmentation (mIoU: 40.21) , density (mIoU:33.53) , and depth + segmentation (mIoU:43.65). The trained models have uploaded on the source code website. Compared to our M2 model, it can show a bias towards segmentation feature.
>
> [Q7] Regarding the cost of the proposed framework.
>
> [A7] Like DADA and CTRL, our batch size is one image. Training time is almost 90 hours for 60K in the first stage, and 10 hours for self-training. The GPU memory usage is about 10GB.
>
> [Q8] Regarding the additional visual illustrations, mathematical notations and abbreviations.
>
> [A8] We have added density and depth maps to Figures 2 and 3 on the source code website. Thanks for your careful review, we will standardize our expression on mathematical notations and abbreviations. But in Line 140,  it is inappropriate to omit the mean and the variance, because N here is not standard normal distribution.
>
> [Q9] Justify the hyper-parameter selections.
>
> [A9] We select the hyper-parameter referring to the similar published work DADA and CTRL, and adjust them by our experiments.
>
> [Q10] The visualization of the feature map (if available).
>
> [A10] Thank you for your references, and we use t-SNE to visualize different joint-space features, which have been uploaded on the website. We will also think about the method you mentioned.
>
> [Q11] Regarding some typos spotted, Figure 1 and some motivation.
>
> [A11] We will fix all these in the final paper.
>
> We gratefully appreciate you for your carefully review and help us to improve the quality of our final paper.

---

### Meta-Review · Area_Chair_veGQ · 2022-08-24

**Recommendation:** Accept
**Confidence:** Certain

**Metareview:**

The paper proposes a depth-aware segmentation framework that leverages unsupervised domain adaptation by segmenting, regressing the depth, and estimating the depth density distribution.  The reviewers' doubts about the paper were addressed during the rebuttal, and the reviewers agree that the authors answered most of their concerns, and that the remaining problems are not an impediment for publication.  While the paper doesn't achieve SotA, the new ideas and the shown experiments are enough to validate the proposal, and its ideas will be of interest to the community.

I recommend the paper for publication given the contributions on the multi-task setup and the mixture of domains, and that the experiments validate the proposal.

**Award:**

No

---

### Decision · Program_Chairs · 2022-09-14

Accept